# The impact of working in academia on researchers' mental health and well-being: A systematic review and qualitative meta-synthesis

Helen Nicholls[1]*, Matthew Nicholls[2], Sahra Tekin[1], Danielle Lamb[3‡], Jo Billings[1‡]

1 Division of Psychiatry, Faculty of Brain Sciences, University College London, London, United Kingdom,
2 MRC Molecular Haematology Unit, MRC Weatherall Institute of Molecular Medicine, Radcliffe Department of Medicine, University of Oxford, Oxford, United Kingdom, 3 Department of Applied Health Research, Institute of Epidemiology and Health Care, University College London, London, United Kingdom

☉ These authors contributed equally to this work.
‡ DL and JB also contributed equally to this work.
* helen.nicholls.20@ucl.ac.uk

**Data Availability Statement:** All relevant data are within the paper and its Supporting Information files.

## Abstract

### Objective

To understand how researchers experience working in academia and the effects these experiences have on their mental health and well-being, through synthesizing published qualitative data.

### Method

A systematic review and qualitative meta-synthesis was conducted to gain a comprehensive overview of what is currently known about academic researchers' mental health and well-being. Relevant papers were identified through searching electronic databases, Google Scholar, and citation tracking. The quality of the included studies was assessed and the data was synthesised using reflexive thematic analysis. The review protocol was registered on PROSPERO (CRD42021232480).

### Results

26 papers were identified and included in this review. Academic researchers' experiences were captured under seven key themes. Job insecurity coupled with the high expectations set by the academic system left researchers at risk of poor mental health and well-being. Access to peer support networks, opportunities for career progression, and mentorship can help mitigate the stress associated with the academic job role, however, under-represented groups in academia are at risk of unequal access to resources, support, and opportunities.

### Conclusion

To improve researchers' well-being at work, scientific/academic practice and the system's concept of what a successful researcher should look like, needs to change. Further high-

**Funding:** This publication was made possible through funding provided by a Bloomsbury Colleges Studentship in partnership with The McPin Foundation (Grant reference: ES/P000592/1). This report is independent research supported by the National Institute for Health Research ARC North Thames. The views expressed in this publication are those of the author(s) and not necessarily those of the National Institute for Health Research or the Department of Health and Social Care. A representative of The McPin Foundation - a mental health research charity - provided feedback on the final draft of this manuscript.

**Competing interests:** The authors have declared that no competing interests exist.

quality qualitative research is needed to better understand how systemic change, including tackling inequality and introducing better support systems, can be brought about more immediately and effectively. Further research is also needed to better understand the experiences and support needs of post-doctoral and more senior researchers, as there is a paucity of literature in this area.

## Trial registration

The review protocol was registered on PROSPERO (CRD42021232480).

## Introduction

The university sector has undergone substantial changes over the last decade [1]. Across the world, universities have become increasingly business-like, often focusing at an institutional level on maximising income streams as opposed to focusing on the original ethos of expanding and training young minds [2]. Indeed, academics are caught up in a range of initiatives which measure job performance including global university league tables, research league tables, and student satisfaction surveys [3–5]. The better the performance across these initiatives, the more revenue a university is likely to attract. These performance rankings can help determine the allocation of public resources and research funding [4], and are often used in the global competition between universities to recruit fee-paying domestic and overseas students [5].

Research output is central to a university's reputation, and critically, largely determines where they fall in the global university rankings [4]. Given the importance placed on research output, it is unsurprising that a good research reputation–often characterized by frequent publication in high impact journals, and a continued ability to successfully obtain research funding–can be critical for career progression in academia [6].

Perceptions of research culture can vary depending on the institution or individual [7, 8]. However, emerging evidence suggests that a large majority of university research cultures are characterized by job insecurity, competing demands in the form of both teaching and research work, long hours including unpaid and uncontracted work, brutal competition amongst peers to succeed in academia, and an immense pressure to publish papers and win research funding [8, 9]. These characteristics have left researchers experiencing high levels of stress, which have the potential to impact negatively on their mental health and well-being [10].

Well-being and mental health are terms which are often used interchangeably, yet research suggests that whilst they should be viewed as linked, they are distinct concepts [11, 12]. Mental health encompasses a spectrum of experience, ranging from good mental health to mental illness. Good mental health is more than just the absence of mental illness, but rather the presence of particular mcontruental skills, habits and capacities [12] that enable an individual to effectively react to, or deal with the environment around them [13, 14]. Well-being on the other hand is a more holistic term, reflecting broader social, physical, and economic experiences. It is often indicative of how closely an individual can live their life in accordance with how they want to. Good well-being is associated with developing robust relationships, reaching individual potential, and being able to engage in activities of personal value and meaning [15].

The high levels of stress inherent in the academic researcher population has been shown to increase their risk of experiencing burnout and depression [16]. Early career researchers, a term often used to describe doctoral researchers and post-doctoral researchers [17, 18], are thought to be particularly at risk of experiencing common mental health difficulties due to the

job precarity that characterizes this career stage [19], and the prevalence of top-down power dynamics which can prevent the disclosure of bullying, harassment, and exploitation [8].

A small, yet growing number of quantitative studies utilizing author-created questionnaires and validated mental health measures have given an indication as to the prevalence and severity of mental-ill health amongst postgraduate researchers in particular. Evans et al., [20] utilised the Generalised Anxiety Disorder questionnaire (GAD-7) and the Patient Health Questionnaire (PHQ-9) to show that postgraduate researchers (comprising both MSc students and doctoral researchers) were six times more likely to report experiencing anxiety and depression compared to those in the general population, with poor work life balance and poor mentor relationships being cited as correlating with worse mental health outcomes. Similarly, a recent systematic review and meta-analysis found that 24% of doctoral researchers displayed clinically significant symptoms of depression and 17% displayed clinically significant symptoms of anxiety, rates which were identified to be similar to estimated prevalence rates in other high stress populations including medical students and resident physicians [21].

Estimates of the prevalence and severity of specific mental health difficulties amongst postdoctoral researchers and more senior researchers are scarce, however, a recently released report by Education Support found that out of 2,046 academic (85.9%) and academic-related staff (14.1%) in the UK, 53.2% showed probable signs of depression [22]. This echoes a recent report by the Wellcome Trust wherein 34% of researchers across multiple career stages (the vast majority located in universities across the globe), stated that during their research career, they had sought professional help for depression or anxiety [8].

Interestingly, Kinman & Johnson [1] have also noted that factors such as secure employment, autonomy, and teamwork which have been shown to protect university employees, including academic staff, against the more stressful aspects of the job, are not as prevalent as they once were in university sectors across the UK, USA and Australia.

Whilst quantitative-focused studies on this topic have brought to light findings that can be considered concerning for the academic community, the work that does exist tends to treat the university workforce (which comprises both academic and non-academic staff) as a homogenous group [16], and they are limited in their ability to capture a researcher's lived experience due to fixed response options. Through using a qualitative research design, responses can be probed and underlying drivers and factors can be uncovered [23], enabling a more in-depth understanding of academic researchers' experiences and how this relates to their mental health and well-being from a self-report perspective (rather than through fixed and often inflexible clinical screening tools or measures). Nevertheless, the qualitative research in this area often tends to focus on examining discrete aspects of mental health, well-being, or the researcher experience [6, 24]. This is unsurprising given the varying ways in which the constructs of mental health and well-being can be conceptualised, and the difficulty in clearly defining the academic researcher population either as a whole, in terms of career stage, or in terms of discipline [14, 25]. This disparity in the existing qualitative literature, however, makes it difficult to ascertain what we currently know about how academic researchers experience working in academia, and the effect these experiences have on their mental health and well-being.

We chose to conduct a systematic review and qualitative meta-synthesis to enable us to identify similarities, contradictions and patterns across existing published data in order to both better understand researchers' experiences and develop new insights into this topic [26]. Integrating data related to this topic area can aide in informing local organisational policy which can better support academic researchers' well-being and can help guide future work in this area.

Through synthesizing existing qualitative data, we aimed to address our research question; how do researchers experience working in academia, and what effect do these experiences have on their mental health and well-being?

## Method

We followed the guidance provided by Lachal et al., [26] on synthesising qualitative literature. The review protocol was registered with PROSPERO, the NIHR's International Prospective Register of Systematic Reviews (registration number: CRD42021232480). PRISMA (Preferred Reporting Items for Systematic Reviews and Meta-analyses) guidance was adhered to throughout this review [27]. Please see supplementary information S1 File for the PRISMA checklist.

### Search strategy

The following electronic databases were searched for relevant academic papers: PsycINFO (Ovid), EMBASE (Ovid), CINAHL Plus, PubMed, SCOPUS, Web of Science.

We searched the databases from inception to January 2021, with an English language restriction (due to limited resources for translation). Key words related to the research question (including 'mental health', well-being, researcher, and qualitative) were organised under the headings of the SPIDER (Sample, Phenomenon of Interest, Design, Evaluation, and Research type) tool and then elaborated upon to include alternative terms, related constructs, and database specific subject headings. The search terms were combined as necessary using the Boolean operators OR and AND.

In order to capture relevant literature not indexed in the electronic databases we also conducted searches on Google Scholar using key terms relevant to the research question such as 'qualitative', 'researcher', 'academia' and 'mental health'. The results were sorted by relevance and no date restriction was applied. The first 200 results were downloaded and imported into the reference management software EndNoteX9, along with the search results from the electronic databases, where duplicates were then removed. For all included papers, we also employed citation tracking. This involved searching the reference lists of the original included paper and searching for papers which cite the original included paper. Forward citation tracking was completed in May 2021. Please see supplementary information S2 File for the full search strategy.

### Eligibility criteria

To be included, peer-reviewed research articles reported in English needed to (a) use a qualitative research design or mixed methods design where qualitative data could be extracted, (b) consist of a sample which clearly identified its population as researchers or individuals with research-related responsibilities (carrying out research, publishing papers, applying for funding), (c) consist of a sample which clearly identifies its population as working in a higher education institution (defined here as an institution which awards degree level certificates or above) and, (d) focus sufficiently on researchers' mental health and well-being experiences—accordingly, papers were only included in the analysis if the aim(s) or research question(s) of the paper involved examining an aspect of mental health or well-being (or both) and aspects of mental health or well-being (or both) formed a significant part of the (qualitative) results output. Any aspect of mental health and/or well-being was eligible for inclusion. Examples of topics related to well-being include work-related stress, psychological or physical well-being, emotional health, life/work satisfaction. Examples of topics related to mental health include resilience, coping, or specific mental health difficulties such as depression or anxiety.

Articles were excluded if: (a) they did not focus sufficiently on researchers' mental health and well-being experiences as detailed above under inclusion criterion (d), (b) they focused primarily on experiences outside of academia, (c) the experiences of researchers who work in higher education institutions could not be extracted, (d) they focused on evaluating a workshop, intervention, or policy change or, (e) the information necessary for the data extraction

phase of the review was not present. Corresponding authors were contacted regarding any missing data necessary for data extraction. Where no response was received in one month, the article was excluded on the basis of this missing information. Whilst research on undergraduate students and masters' students were excluded, literature concerning doctoral researchers was eligible for inclusion, as much of the research which explores the experiences of early career researchers has often included doctoral researchers within that paradigm, and they have been noted as playing a key role in the research productivity of universities [17, 18]. Quantitative studies were excluded as their ability to address lived experience is limited [23].

## Data extraction and analysis

The following data were extracted from the eligible papers by the first author (HN): (1) title of the research, (2) author (year) and country, (3) sample size, (4) identifying features of the participants, (5) the aspect(s) of mental health or well-being explored as part of the research aim/question (6) method of qualitative data collection and, (7) method of qualitative data analysis.

The included papers were exported into NVivo Pro version 12 to facilitate analysis. Qualitative data contained within the included papers under headings (such as) 'results' or 'findings' were analysed. The qualitative data analysed included themes, participant quotes, or author interpretations/explanations.

The analytical method used to synthesize the qualitative data in this review was reflexive thematic analysis. Reflexive thematic analysis can be used to explore questions pertaining to participants' experiences and is well suited to answer the research question of this systematic review [28]. The primary author (HN) followed the 6-phase process as outlined by Braun et al., [28], although the analysis was a recursive process and involved movement back and forth between each phase. Following initial immersion in the data through reading and re-reading the papers, the research team comprising HN, JB, DL, ST, and MN, collaboratively developed a provisional coding frame based on eight papers which were identified as being recent enough to cover current practices in academia (studies conducted within the last 5 years at the time of our analysis (2021)), and diverse in terms of examining different aspects of mental health, well-being, and the researcher experience. As the analysis progressed to include the remaining papers, the coding frame was further refined and extended as necessary. Given the exploratory nature of the research question, the process of data coding and theme development was inductive.

When approaching the data, we adopted a critical realist position. That is, we assumed that the qualitative data analysed from the included papers was informative of an objective 'true' reality shared by academic researchers, but we also acknowledge that each individual researcher will have a subjective reality which is mediated by their own perceptions, experiences, and beliefs. Identifying key similarities/themes across the dataset (whilst accounting for contradictory findings) helped us to explore this dual reality.

## Reflexivity

The researcher plays an active role in qualitative research (influencing not only how the data is collected, but also how the data are interpreted through their own personal experiences, knowledge, and assumptions). As such, it is important to comment on the position and composition of the research team, so that the reader is able to come to their own conclusions about the validity and trustworthiness of the analysis produced.

The research team differed in terms of discipline, career stage, gender, and cultural background, which helped to ensure that any personal assumptions or 'blind spots' with regards to assessing eligibility, assessing quality, interpreting the results, or the topic as a whole, were

minimised. Both HN and ST are current doctoral researchers with previous experience in conducting qualitative research, and experience as assistant psychologists. JB is a Consultant Clinical Psychologist and Associate Clinical Professor, and DL is a Senior Research Fellow, both of whom have extensive experience in conducting qualitative research and systematic reviews. MN is a research assistant in the field of molecular medicine. Together, the research team brought a variety of different perspectives and experiences to the development of the synthesis, and this piece of research as a whole.

# Results

## Screening outcome

Fig 1 depicts a PRISMA flow diagram concerning the process of identification, screening, and selection of papers for inclusion [27]. 13,778 articles were identified through searching the bibliographic databases and Google Scholar. Eight additional articles were identified through

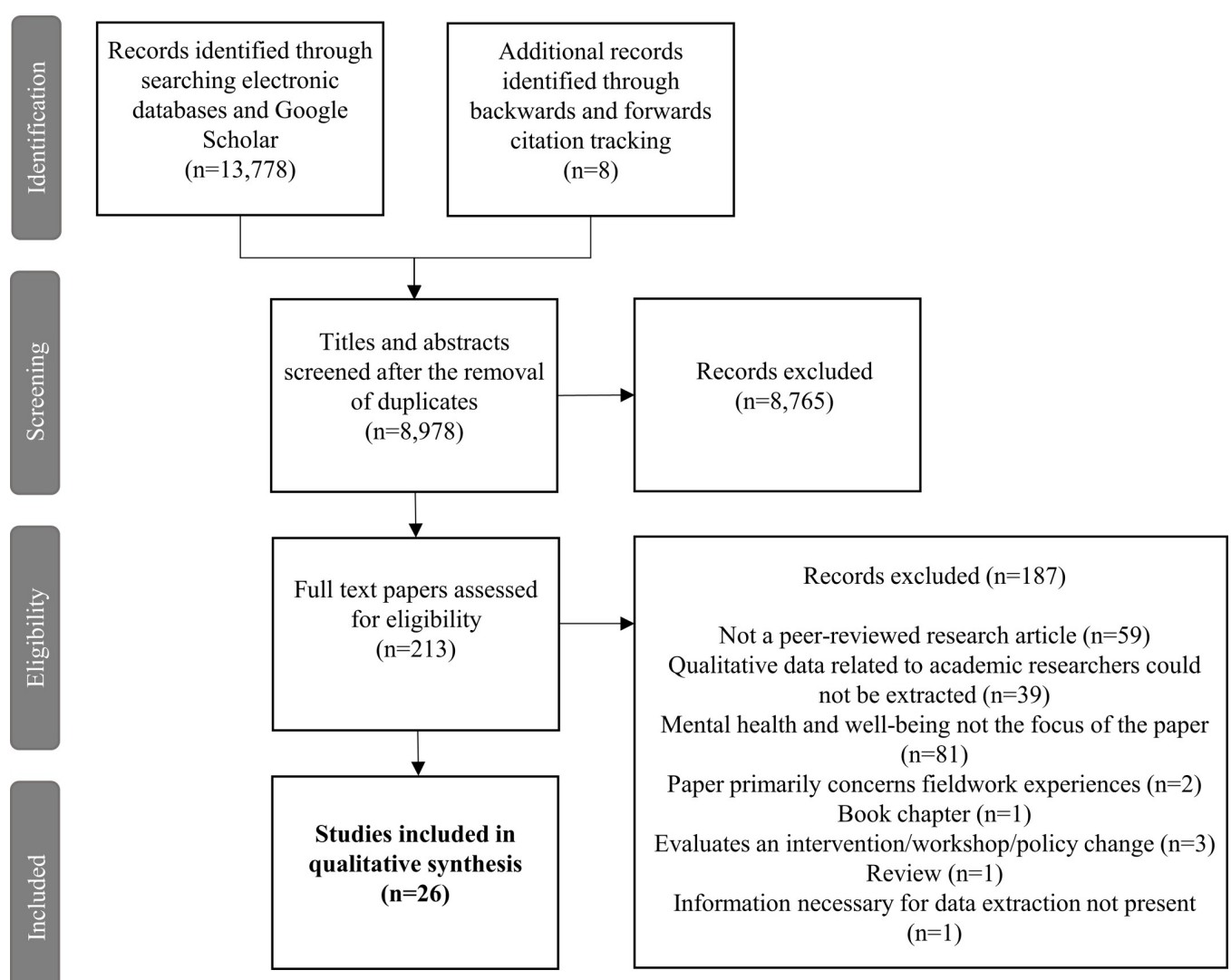

**Fig 1. PRISMA diagram reporting the process of identification, screening, and selection of papers for inclusion in the review.**

citation tracking. Following de-duplication, 8,978 titles and abstracts were screened for relevance by the primary reviewer (HN) using the software tool Rayyan QCRI. Just over 10% (n = 905) of the titles and abstracts were screened independently by a second reviewer (MN). Of the 8,978 papers 8,765 were excluded for irrelevance, leaving 213 full text articles to be sourced and read in full.

All 213 articles were screened for relevance independently by the primary and secondary reviewer. At this stage 187 articles were excluded, for not being a peer-reviewed research article (n = 59), for not focusing on mental health and well-being (n = 81), and for not having extractable qualitative data related to academic researchers (n = 39). Articles were also excluded on the basis of not having the information necessary for data extraction (n = 1), for focusing primarily on fieldwork experiences (n = 2), focusing on evaluating a workshop, intervention, or policy change (n = 3), and finally for being a book chapter (n = 1), or review (n = 1). Any disagreements over eligibility at either the title and abstract stage or the full text stage were resolved through discussion between HN and MN. Where eligibility remained unclear, JB and DL were consulted and a decision was made.

In the protocol submitted to PROSPERO, we outlined our intentions to include relevant grey literature and commentaries written in first person where all necessary data could be extracted. This was to ensure we captured sufficient data for a meta-synthesis to be feasible. However, due to the adequate amount of peer-reviewed research articles available for inclusion in the review, the decision was made to exclude any grey literature and commentaries at the full text screening stage. Attempts were made to find peer-reviewed versions of this literature, which were then screened for relevance instead. Articles which have gone through a peer review process are likely of a good quality [29], which can help to create confidence in the findings. By including peer-reviewed research articles only in our systematic review, we therefore aim to increase confidence in the quality of our findings also.

## Characteristics of the included studies

Of the 26 papers included in the meta synthesis, five papers included participants based in North America (Canada, USA, Mexico), 13 papers included participants from Europe (UK, Finland, Germany, Sweden, Netherlands, Spain), one paper included participants from Asia (China), and nine papers included participants based in Australia and Oceania (Australia and New Zealand). The methods of data collection employed included: surveys with open-ended questions (n = 7), interviews (n = 14), focus groups (n = 4), and autoethnographic excerpts (n = 4). Whilst all participants were associated with conducting research, they varied in terms of career stage and role title. The most common group was that of doctoral researchers, with 14 papers including them as participants. The academic disciplines represented across the papers varied extensively, as did the aspects of mental health and well-being examined. All of the studies were published between 2011 and 2021. Further details pertaining to the characteristics of the included studies can be found in Table 1. The details contained in Table 1 refer to the qualitative component of the papers, where papers are mixed methods or report on more than one study.

## Quality appraisal

Research into the mental health and wellbeing of researchers in academia is still in the early stages, with limited literature published so far. As such, no paper was excluded from this review due to quality limitations. However, each study was given a quality rating through the use of the Critical Appraisal Skills Program (CASP) checklist. The Critical Appraisal Skills Program (CASP) checklist was used to assess the quality of the included studies as it is a frequently

**Table 1. Characteristics of the included studies and quality appraisal outcomes.**

| Author (year) Country | Title | (Participant characteristics) Population studied (%/no. of participants) Academic discipline(s)/ field of study Sex/Gender (%/no. of participants) | Sample size | Aspect(s) of mental health and/or well-being explored | (Study design) Method of data collection Method of data analysis | CASP quality appraisal outcomes (T = totally met, P = partially met, N = not met) | | | | | | | | | |
|---|---|---|---|---|---|---|---|---|---|---|---|---|---|---|---|
| | | | | | | Q1 | Q2 | Q3 | Q4 | Q5 | Q6 | Q7 | Q8 | Q9 | Q10 |
| Barry et al., (2018) [30] Australia | Psychological health of doctoral candidates, study-related challenges, and perceived performance | Doctoral researchers (n = 81) Discipline not stated Sex/gender not stated clearly, however, the sample was stated as being dominated by female participants | 81 | Psychological distress | Survey questionnaire (included open ended questions) Abductive analytical approach | T | T | T | T | T | N | T | T | T | T |
| Berry et al., (2020) [31] UK | Hanging in the balance: Conceptualising doctoral researcher mental health as a dynamic balance across key tensions characterising the PhD experience | Doctoral researchers (n = 12) Science, Arts and Humanities, Social Science Male (n = 9), Female (n = 23) | 32 | Mental health and mental health problems | Focus groups Thematic analysis | T | T | T | T | T | T | T | T | T | T |
| Campbell (2018) [32] UK | Reconstructing my identity: An autoethnographic exploration of depression and anxiety in academia | An academic (n = 1) Law Sex/gender not stated | 1 | Mental health illness | Diary entries Evocative autoethnography | T | T | T | T | T | T | P | T | T | P |
| Chan et al., (2021) [33] Australia | The battle-hardened academic: an exploration of the resilience of university academics in the face of ongoing criticism and rejection of their research | Professor (n = 4), Associate Professor (n = 2), Senior Lecturer (n = 5), Lecturer (n = 1) Health Sciences Male (n = 4), Female (n = 8) | 12 | Resilience | Semi-structured interviews Thematic analysis | T | T | T | T | T | T | T | T | T | T |
| Chubb et al., (2017) [34] UK & Australia | Fear and loathing in the academy? The role of emotion in response to an impact agenda in the UK and Australia | Mid-senior career academics, specifically those who have been Principal or co-Investigators on grant applications) (n = 51) Arts and Humanities, Social Sciences, Engineering and the Physical Sciences, Life and Natural Sciences. Male (n = 31), Female (n = 20) | 51 | Emotion | Semi-structured interviews Thematic analysis | T | T | T | T | T | N | T | T | P | P |
| Cornwall et al., (2019) [35] New Zealand | Stressors in early-stage doctoral students | Early-stage doctoral researchers (n = 152) Discipline not stated Sex/gender not stated | 152 | Stress | Online questionnaire (included two questions which required free-text responses) Thematic analysis | T | T | P | T | P | P | T | T | T | T |
| Cotterall et al., (2013) [36] Australia | More than just a brain: emotions and the doctoral experience | International doctoral researchers (n = 6) Science, Human Science, Business and Economics Male (n = 3), Female (n = 3) | 6 | Emotion | Multiple interviews conducted over a 2-year period (x3 interviews per year) The data was analysed through the lens of activity theory, and involved identifying all emotion-related episodes using linguistic, non-linguistic, and contextual cues | T | T | T | T | T | P | N | T | T | P |

*(Continued)*

**Table 1.** (Continued)

| Author (year) Country | Title | (Participant characteristics) Population studied (%/no. of participants) Academic discipline(s)/ field of study Sex/Gender (%/no. of participants) | Sample size | Aspect(s) of mental health and/or well-being explored | (Study design) Method of data collection Method of data analysis | CASP quality appraisal outcomes (T = totally met, P = partially met, N = not met) | | | | | | | | | |
|---|---|---|---|---|---|---|---|---|---|---|---|---|---|---|---|
| | | | | | | Q1 | Q2 | Q3 | Q4 | Q5 | Q6 | Q7 | Q8 | Q9 | Q10 |
| Darabi et al., (2017) [37] UK | A qualitative study of the UK academic role: positive features, negative aspects and associated stressors in a mainly teaching-focused university | Associate Lecturers (n = 2), Lecturers (n = 6), Senior Lecturers (n = 16), Principal Lecturers (n = 5), Professors (n = 2). Discipline not stated. Male (n = 12), Female (n = 18) Transgender (n = 1) | 31 | Coping | Structured interviews (online) Thematic analysis | T | T | T | T | T | P | T | T | T | T |
| Herbert et al., (2014) [6] Australia | The impact of funding deadlines on personal workloads, stress, and family relationships: a qualitative study of Australian researchers | Early career researchers (e.g., Assistant Lecturer, Lecturer) (26%) mid-career researchers (e.g., Senior Lecturer) (27%), senior level researchers (e.g., Associate Professor, Professor) (39%), role not stated (8%). The target group was researchers with experience of applying for a NHMRC Project Grant. Discipline not stated Sex/gender not stated | 215 | Stress | Online survey (included an open-ended question) Thematic analysis | T | T | T | T | T | N | P | T | T | T |
| McGee et al., (2019) [38] USA | "I Know I Have to Work Twice as Hard and Hope That Makes Me Good Enough": Exploring the Stress and Strain of Black Doctoral Students in Engineering and Computing | Black doctoral researchers (n = 38), postdoctoral researchers (n = 3), recently awarded PhD/graduated (n = 1), role not stated (n = 6) Engineering and Computing fields Male (n = 29), Female (n = 19) | 48 | Stress and coping | Semi-structured interviews & focus groups Transcendental phenomenology | T | T | P | P | T | T | T | T | T | T |
| Medina et al., (2016) [39] Mexico | Emotional Burnout Syndrome in Women Researchers: The case of the Juarez Autonomous University of Tabasco | Researchers with symptoms of Emotional Distress Syndrome (EDS) (n = 13) Discipline not stated. Female (n-13) | 13 | Emotional Distress Syndrome | Semi-structured interviews Open, axial, and selective coding | T | T | T | T | T | T | N | T | T | T |
| Muir et al., (2021) [40] North America & Australia | Examining Professional Development among Faculty Members across Varying Career Stages in Kinesiology | Assistant Professor (n = 1), Associate Professor (n = 1), Professor (n = 1) Kinesiology Sex/gender not stated | 3 | Coping | Semi-structured phone interviews Abductive analysis | T | T | P | T | T | N | P | P | T | T |
| Nikischer, (2019) [41] USA | Vicarious trauma inside the academe: understanding the impact of teaching, researching, and writing violence | Tenure-track faculty member (n = 1) Field of violence against women Sex/gender not stated | 1 | Vicarious and secondary trauma | Reflective journals Analytic autoethnography (not explicitly stated) | T | T | T | T | T | P | N | T | T | T |
| Pappa et al., (2020) [42] Finland | Sources of stress and scholarly identity: the case of international doctoral students of education in Finland | International doctoral researchers (n = 11) Educational Sciences Male (n = 1), Female (n = 10) | 11 | Stress | Semi-structured interviews Thematic analysis | T | T | P | T | P | T | T | T | T | T |

(*Continued*)

**Table 1.** (Continued)

| Author (year) Country | Title | (Participant characteristics) Population studied (%/no. of participants) Academic discipline(s)/ field of study Sex/Gender (%/no. of participants) | Sample size | Aspect(s) of mental health and/or well-being explored | (Study design) Method of data collection Method of data analysis | CASP quality appraisal outcomes (T = totally met, P = partially met, N = not met) | | | | | | | | | |
|---|---|---|---|---|---|---|---|---|---|---|---|---|---|---|---|
| | | | | | | Q1 | Q2 | Q3 | Q4 | Q5 | Q6 | Q7 | Q8 | Q9 | Q10 |
| Schmidt et al., (2014) [43] Sweden | Experiences of well-being among female doctoral students in Sweden | Doctoral researchers (n = 12) Biology, Business Administration, Health Sciences, Nursing, Informatics, Public Health. Female (n = 12) | 12 | Well-being | Focus groups Structural analysis | T | T | T | T | T | T | T | T | T | T |
| Skakni et al., (2017) [44] UK | Post-PhD researchers' experiences: an emotionally rocky road. | Post-PhD researchers (n = 71) Social sciences, Humanities, Education, Life Science, Health Sciences and Engineering Male (35%), Female (65%) | 71 | Emotion | Online survey (included open ended questions) & semi-structured interviews Four-step iterative process inspired by thematic analysis | T | T | T | T | T | P | P | T | T | T |
| Stubb et al., (2011) [45] Finland | Balancing between inspiration and exhaustion: PhD students' experienced socio-psychological well-being | Doctoral researchers (n = 383) Humanities, Medicine, Behavioural Sciences Male, Female (numbers not stated for the qualitative component) | 383 | Socio-psychological well-being. | Online survey (included open ended questions) Content analysed using an abductive strategy | T | T | P | P | P | N | N | T | T | T |
| Todd, (2021) [46] UK | Experiencing and embodying anxiety in spaces of academia and social research | Doctoral researcher (n = 1) Social Sciences (Human Geography). Male (n = 1) | 1 | Anxiety | Autoethnographic excerpts from field notes Analytic autoethnography (not explicitly stated) | T | T | P | T | T | P | N | T | T | T |
| van der Weijden et al., (2016) [47] Netherlands | Career satisfaction of postdoctoral researchers in relation to their expectations for the future | Postdoctoral researchers (n = 21) Sciences (e.g., Chemistry, Mathematics, Computer Science, Astronomy, Physics, Biology, Environmental Science), Technical Sciences and Engineering, Humanities, Social and Behavioural Sciences, Archaeology Male (n = 134), Female (n = 91) | 225 total respondents (the number of respondents who contributed to the qualitative component is unknown) | Job satisfaction | Online survey (includes open-ended questions) Open coding | T | T | T | T | T | P | N | P | N | T |
| Waight et al., (2018) [48] UK | Doctoral students' access to non-academic support for mental health | Doctoral researchers (n = 594) Discipline not stated Online survey (sex/ gender not stated), Focus group (Male n = 11, Female n = 24) | Online survey (559 total respondents–the number of respondents who contributed to the qualitative component is unknown) Focus group (35) | Mental health | Online survey (included open comments section) & focus groups Coding using an inductive, interpretative approach | T | T | T | T | T | T | P | T | T | T |
| Wang et al., (2019) [49] China | Towards the contributing factors for stress confronting Chinese PhD students | Doctoral researchers (n = 10) Humanities and Social Science Male (n = 5), Female (n = 5) | 10 | Stress | Semi-structured interviews Open, axial, and selective coding | T | T | T | T | T | T | T | T | T | T |

(Continued)

**Table 1.** (Continued)

| Author (year) Country | Title | (Participant characteristics) Population studied (%/no. of participants) Academic discipline(s)/field of study Sex/Gender (%/no. of participants) | Sample size | Aspect(s) of mental health and/or well-being explored | (Study design) Method of data collection Method of data analysis | CASP quality appraisal outcomes (T = totally met, P = partially met, N = not met) | | | | | | | | | |
|---|---|---|---|---|---|---|---|---|---|---|---|---|---|---|---|
| | | | | | | Q1 | Q2 | Q3 | Q4 | Q5 | Q6 | Q7 | Q8 | Q9 | Q10 |
| Weise et al., (2020) [50] Spain | Significant events and the role of emotion along doctoral researcher personal trajectories | Doctoral researchers (n = 10) Social Sciences Female (n = 7), sex/gender not stated (n = 3) | 10 | Emotion | Multimodal, semi-structured interviews Qualitative interpretative approach of a transversal nature | T | T | T | T | T | P | T | T | T | T |
| White, (2018) [51] Australia | Are New Career Models for Science Research Emerging | Research scientists including doctoral researchers in the third year of their candidature, early post docs, mid-career postdocs, and senior Principal Investigators/lab heads/divisional heads (numbers not stated) Sciences Male, Female (number not stated) | 40 | Job satisfaction | Interviews Qualitative content analysis | T | T | T | T | T | T | T | T | T | P |
| Yang et al., (2020) [52] Australia | Psychological Adjustment of Chinese PhD Students: A Narrative Study | Chinese doctoral researchers based in Australia (n = 6) Marketing, Education, Management, Linguistics, Communications, Economics Male (n = 1), Female (n = 5) | 6 | Stress-coping strategies | Narrative inquiry method (3 stages of individual interviews) Open, axial, and selective coding | T | T | T | T | T | P | N | T | T | T |
| Young et al., (2017) [53] Canada | Women Reflect on Being Well in Academia: Challenges and Supports | Academics–on contract (n = 3), tenure track (n = 5), tenured (n = 5). Education, Social Studies, Counselling Psychology Female (n = 13) | 13 | Health and wellness | Personal narratives Thematic analysis | T | T | T | P | P | N | N | T | T | P |
| Ysseldyk et al., (2019) [54] Canada & Germany | A Leak in the Academic Pipeline: Identity and Health Among Postdoctoral Women | Postdoctoral researchers (n = 21) Psychology, Physics, Political Science, Natural Sciences. Female (n = 21) | 21 | Mental health | Semi-structured interviews Thematic analysis | T | T | T | T | T | N | T | P | T | T |

used instrument that is recommended by the Cochrane Collaboration, and it addresses many of the principles underlying qualitative research [55].

We used a three-point scale which is advocated by Lachal et al., [26] to classify criteria as–totally met, partially met, and not met. The criteria were as follows, Q1: Was there a clear statement of the aims of the research? Q2: Is a qualitative methodology appropriate? Q3: Was the research design appropriate to address the aims of the research? Q4: Was the recruitment strategy appropriate to the aims of the research? Q5: Was the data collected in a way that addressed the research issue? Q6: Has the relationship between researcher and participants been adequately considered? Q7: Have ethical issues been taken into consideration? Q8: Was the data analysis sufficiently rigorous? Q9: Is there a clear statement of findings? Q10: How valuable is the research?

The studies were independently assessed on quality by HN and ST. Any disagreements were resolved through discussion. Individual study quality ratings can be found in Table 1.

## Meta-synthesis

We identified seven key themes through the reflexive thematic analysis. The key themes along with their corresponding sub-themes (which are highlighted in bold and italicised in the text) are reported below with illustrative extracts. Fig 2 visually depicts the interconnected nature of the main themes and sub-themes.

**Insecurity and career prospects.** Issues with *financial insecurity* spanned researchers' experiences across countries, disciplines, and career stages, and often resulted in feelings of worry and stress. Researchers from the UK commented on a scarcity of funding to effectively support students at undergraduate level and above [37], whilst professors at varying levels from North America and Australia commented on a lack of funding to support professional development activities [40]. Doctoral researchers' financial concerns centred mainly on the financial limits imposed by the scholarship or stipend they receive throughout the duration of their study:

'. . . receiving *"a stipend that can barely [. . .] support your living'* as a doctoral student is not the same as other people earning money, like, real money by working" (Doctoral researcher, Educational Sciences, Finland—[42]).

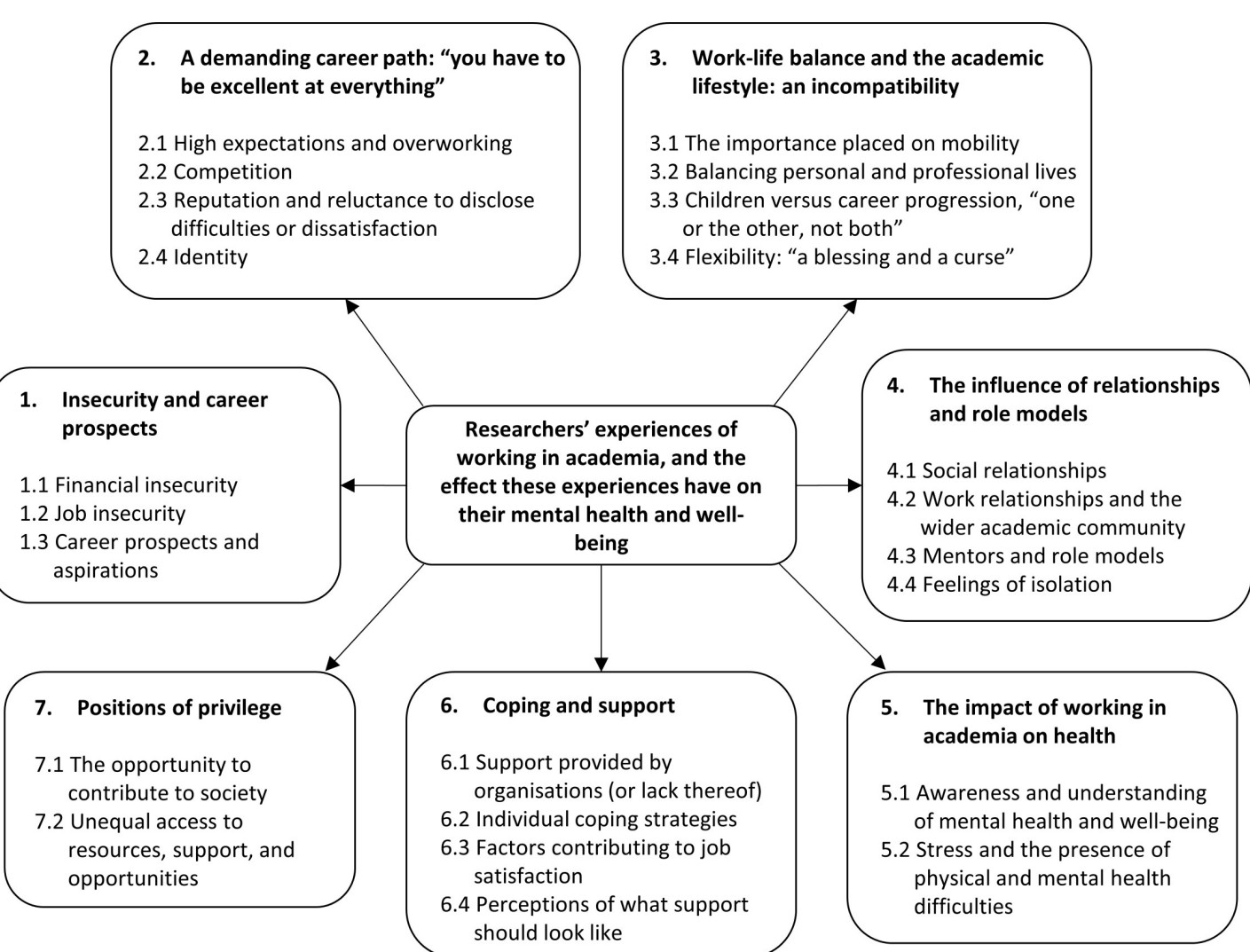

**Fig 2. A visual representation of the main themes and sub-themes identified through the meta-synthesis.**

For many post-doctoral researchers and those in the later stages of their career, economic precarity was also linked directly to *job insecurity*. Researchers from Australia in the later stages of their career [6, 51] drew attention to the importance of successfully obtaining a grant to fund research activity, which helps to maintain both current job contracts and research personnel. The likelihood of ongoing employment being dependent on the outcome of a funding round or securing a grant–the process of which was not always considered fair–placed extensive pressure on researchers, which ultimately impacted negatively on their well-being:

*". . .the chance of anyone with even a modicum of expertise in your field reviewing your grant is basically zero"* (Mid-career researcher, Australia—[6]).

*". . .Many people anxiously await the grant outcome to see if they are out of work in six weeks"* (Senior researcher, Australia—[6]).

The stressful nature of precarious work contracts was further compounded by a lack of communication on behalf of the universities around contract renewal:

*". . . my contract was coming up for renewal, and my university just messed me around. . .they weren't telling me anything. . ."* (Post-doctoral researcher, Education, UK—[44]).

Others felt that this precariousness also prevented researchers from expressing dissatisfaction with current work practices at an institution:

*". . .you can lose your job if you question practices of a higher level. . .."* (Professor, Engineering Education, Australia—[34]).

Both financial and job insecurity impacted on researchers' *career prospects and aspirations*. Unsurprisingly, perceptions of career prospects were influenced by the reliability of a funding bridge between research projects or posts. However, they were also influenced by the lack of "tenure-track" [54] and permanent positions available in academia. Indeed, some researchers felt that the unpredictability of securing a permanent position de-valued their achievements, hard-work, and expertise:

*". . . it is not weird. . .to expect after such long studies and with such a great CV, to get a permanent position as reward and acknowledgement. . ."* (Post-doctoral Researcher, Netherlands—[47]).

Nevertheless, despite restricted career prospects which may compromise mental health and well-being, a number of researchers, particularly those in the post-doctoral stage, stated that it was their intention to remain in academia. Doctoral researchers, however, were noticeably more hesitant in committing to a single career path:

*"I am green with envy and stressful when I see my classmates at college are well-settled down in their career while I am still struggling for a PhD, my career still being an illusion"* (Doctoral researcher, China—[49]).

It is important to keep in mind that limited access to career workshops or coaching [47] and thus limited preparations for a non-academic career path, may influence a researcher's intent to remain within academia:

*". . . I feel I have little to offer outside of the university sector and am unsure what I could realistically go for!"* (Participant 12, UK—[37]),

**A demanding career path: "you have to be excellent at everything".** The sub-theme *high expectations and overworking* was present across all 26 of the included papers. High expectations encapsulates the pressure to engage with the three domains of research, teaching, and service (for example conducting a review of a program [53]), whether trained in these domains or not, the pressure to handle competing demands with strict deadlines, to work unpredictable and long hours, to be independent, to handle multiple counts of criticism and rejection [33], and to be resilient in the face of these expectations. Ultimately, researchers were expected to be focused on impact and productivity [46] rather than individual/researcher well-being or emotional capacity.

These expectations were set by the system, and consequently by colleagues and the researcher themselves. Prominent factors which made meeting these expectations difficult, particularly for post-doctoral researchers and researchers in the later stages of their career included the introduction of new research policies which were not conducive to all disciplines, increased student numbers without the necessary resources in place to manage this, increased administrative loads, and the expectation to provide pastoral support to students. The expectation to produce research that is 'impactful' appeared to more negatively impact the well-being of those from less applied/theoretical disciplines [34], whilst the expectation to provide pastoral support was noted as falling more heavily on female researchers:

*". . . she had people queuing out the door for office hours even if they weren't really her students . . ."* (Doctoral researcher, Arts & Humanities, UK—[31]).

To meet the job demands of academia and/or gain the skills and accolades necessary to secure an ever-elusive permanent position, researchers across countries, career stages, and disciplines described the extra work they took on and the long hours they worked, along with the productivity guilt that ensued when they were not able to meet their own or others' expectations. This left them at risk of stress and burnout:

*". . .if it's like 4 pm and. . .my experiment hasn't worked, immediately my brain is like "Well you should start it again and leave work at 10pm. . .finish it, get it right. . ."* (Doctoral researcher, Science, UK—[31]).

Research cultures characterised by high expectations, job precarity, and reduced opportunities for permanent positions engendered a sense of *competition* among the research community often described as '*nasty, aggressive and unpleasant*' [51]. Nevertheless, it was difficult not to perpetuate this sense of competition, so as not to feel at a disadvantage career-wise:

*". . . I will still secretly judge if somebody always goes home at 4pm, and I know I shouldn't. . . But there is this. . . highly competitive spirit that everybody sort of expects, that if you want to be the best then you have to work 80 h a week . . ."* (Post-doctoral researcher, Canada—[54]).

In the interest of career progression, both early career researchers and researchers at more senior levels expressed wanting to maintain a *reputation* of being able to meet the high expectations set:

*'. . . I have known academics who have hidden their mental distress for fear of being pigeon-holed as flimsy and undependable'* [32].

Interestingly, career stage did not appear to impact researchers' **reluctance to disclose difficulties or dissatisfaction** (that is, difficulties of an academic, mental health, or well-being related nature):

*". . .I don't want to give the impression that I'm already failing"* (Doctoral researcher, Science, UK—[31]).

Due to this general reluctance to disclose, researchers were left at risk of blaming any difficulties or dissatisfaction associated with their job on themselves. This risk was further compounded by facing, or expecting to face, negative reactions from colleagues or supervisors when difficulties or dissatisfaction related to the workplace were shared:

*"I just felt like. . . they wouldn't listen to me as a person and they would just say, "Hey, see these Black kids can't cut it""* (Post-doctoral researcher, Mechanical Engineering, USA—[38]).

For doctoral researchers specifically, the reluctance to disclose was also related to confusion over the extent to which the supervisory relationship is pastoral, unknown limits to confidentiality, and a fear of overburdening others:

*". . .they're stressed and it feels like a lot to say to them. . .'Can we talk to you more. . .?"* (Doctoral researcher, Arts & Humanities, UK—[31]).

It is important to note that some researchers encountered both understanding and help from their colleagues and supervisors following the disclosure of difficulties [31, 32]. However, the extent to which this understanding lasted was limited in some cases:

*'. . .two focus group participants with chronic mental health conditions stated that although supervisory teams were sympathetic when they first learned of the student's condition, participants felt that this was soon forgotten or dismissed with the expectation that they must surely be 'over it' after a period of time'* [48].

Ultimately, being perceived as meeting or not meeting the expectations set by themselves, colleagues, or the system as a whole may have an impact on a researchers' confidence in their ability to do their job, not only affecting their well-being, but also their sense of **identity** as an academic, and thus their sense of belonging to the academic community.

*Receiving an award from a research society based on my presentation and work [. . .] I felt that I was recognised as an experienced researcher who could convey my research and was becoming an expert in my field"* (Post-doctoral researcher, Oncology, UK—[44]).

Indeed, some researchers commented on not feeling like their current self is living up to the perception of the 'proper' or 'ideal' academic. These feelings of self-doubt and inadequacy (despite evidence to the contrary) are indicative of imposter syndrome, a well-known phenomenon in academia that can impact negatively on mental health:

*'In some cases, women also explicitly attributed mental health issues to imposter syndrome, as in the case of a Canadian postdoc who reported:* "*Mentally I think definitely there's been some bouts of depression. You know, definitely some imposter syndrome. . . So with that, you know, definitely some anxiety. . .*" [54].

A factor which further impacted doctoral researchers' identity and sense of belonging at work was uncertainty around whether they are a member of the student body or faculty:

"*. . .we are like ghosts in the campus. We are part of the faculty, but we are not*" (Doctoral researcher, Arts & Humanities, UK—[31]).

**Work-life balance and the academic lifestyle: An incompatibility.** The high expectations set by academia coupled with **the importance placed on mobility** for continued employment: "*I'm working in a city for 2 years and then I'm expected to move to a whole new country. . .*" (Post-doctoral researcher, Germany—[54]), networking, and career progression: "*One of the requirements for the fellowship above my level specifically says you have had to work overseas*" (Participant, Sciences, Australia—[51]) often made **balancing personal and professional lives**, difficult. Researchers across career stages, academic disciplines, and countries described academia as inherently inflexible in this regard:

"*. . .You are either expected to play the game in full or get out*" (Post-doctoral researcher, Canada—[54]).

Feeling unable to take breaks and engage with other meaningful activities had the potential to lead to high levels of stress and burnout. Feelings of frustration and guilt were also particularly prevalent in responses which mentioned conflict between job demands and family systems:

"*. . .my family is the most important to me, but so is my career, and there is when I go into conflict. . .I do not want to leave either of them but I cannot be in both places at the same time. . .*" (Participant, Mexico—[39]).

Indeed, a key stressor spoken about primarily by female researchers, was that of when to start a family. Early career researchers in particular described the following tension: **children versus career progression, "one or the other not both".** Many perceived that having children was generally stigmatized and discouraged within the academic environment due to the potential loss of productivity:

"*The gossip in my department was that. . . the climate was not very conducive for women to become pregnant . . . they become less useful for the department during their time off. . .*" (Post-doctoral researcher, Germany—[54]).

Ultimately, in the context of applying for funding [51], publishing [54], and the academic job market as a whole; pregnancy, taking maternity leave, and raising a child left female researchers feeling at a disadvantage compared to their peers without children:

"*. . . I feel like that does harm your career. Because I don't think it's recognized. . . you're still expected to be producing a certain number of publications even if you are taking time off to have kids. . .*" (Post-doctoral researcher, Canada—[54]).

Nevertheless, flexible working hours and flexibility in terms of idea development appeared to be associated with good well-being as it allowed academic researchers to retain a certain level of autonomy and independence over their personal and professional lives. However, the tension, *flexibility: "a blessing and a curse"* was prevalent in the literature. Some early career researchers including doctoral researchers and post-doctoral researchers, associated this flexibility with difficulties in maintaining motivation and feeling a sense of achievement:

*"The flexibility, it's both a blessing and a curse really, every day you kind of plan for yourself, and it's a blank slate. But admittedly a lot of times I wake up and I'm not sure what I'm going to achieve that day and I don't achieve anything"* (Post-doctoral researcher, Canada—[54]).

**The influence of relationships and role models.** Researchers' *social relationships* were often described as being sources of support. Social relationships were described as protective against the experience of mental health or well-being-related difficulties, and often aided in maintaining a good work life balance:

*". . .my husband and children allowed me to stay sane because it forced me to make time for other things than work"* (Professor, Kinesiology—[40]).

However, social relationships could also be a source of stress. Due to the demands and high expectations associated with the academic job role, establishing, or maintaining social relationships outside of academia could be complex:

*"Only the strongest relationships survive. . .I focus on only the closest family members [for] maintaining relationships. Other relationships have had to adapt. . .or, more often, disintegrate"* (Senior researcher, Australia—[6]).

Researchers' also spoke about their ***work relationships and the wider academic community*** in the context of being protective against the demands of the job. Positive interactions at work helped to combat feelings of loneliness, isolation, and mental distress. Receiving peer support from those within the same discipline, those at a similar career stage, or those with other similar personal characteristics was considered particularly beneficial, as it led to a feeling of 'togetherness' and a sense of community:

*". . . there is no one else that understands you as well as another doctoral student . . ."* (Doctoral researcher, Sweden—[43]).

Nevertheless, researchers work relationships could also be stressful. The competition to get ahead in academia can encourage both negative self-comparisons and fractious relationships to form between colleagues, which imposed limits on opportunities for peer support:

*"Colleagues take advantage. . .It made me understand the kind of person I find diffcult to work with"* (Post-doctoral researcher, Medical Sciences, UK—[44]).

For researchers who also taught, their interactions with students were described as similarly double edged, being both a source of job satisfaction and stress:

*"There is a lack of respect with some. They disrupt lectures and send quite rude emails demanding attention"* (Participant 28, UK—[37]).

Researchers from all career stages spoke about the importance of **mentors and role models**, that is, having someone to guide them throughout their career as a researcher. For doctoral researchers, their supervisor was seen as somebody who could contain worries, strengthen the doctoral researcher's confidence, and increase motivation. However, supervision could negatively impact mental health and well-being if it was perceived as not meeting the doctoral researchers' own needs and expectations or was considered unhelpful or harmful. A lack of formal rules and training were thought to encourage negative supervisory practices:

> *"[supervisors] might be amazing scientists, but they have never been trained in. . . managing collective people"* (Doctoral researcher, Science, UK—[31]).

Closely tied to supervision and mentorship was the idea of a role model, that is, a person who is looked up to by others as an inspirational example to be imitated. Role models were particularly important for both women and those from a Black ethnic background, who are under-represented among the senior levels in academia. This lack of representation at the higher levels led to feelings of not belonging amongst early career researchers who share these characteristics:

> *". . .I feel like engineering in general is much harder for minorities because they don't have a lot of people they can look up to. . ."* (Doctoral researcher, Computer Science, USA—[38]).

The absence of either good supervision, role models, or peer or social networks led to **feelings of isolation** and loneliness, which had the potential to significantly impact not only researchers' mental health and well-being, but also their productivity at work [41]. In the context of this review, particular groups at risk of isolation included female researchers [54], researchers from a Black Ethnic background [38], and part-time or international doctoral researchers [48]:

> *". . .you end up being totally isolated and I think it's easier to some extent for British or when you have your family because even if they don't know anything what you're doing they are still there to support you. . ."* (Doctoral researcher, UK—[48]).

**The impact of working in academia on health.** Researchers' **awareness and understanding of mental health and well-being** varied across the included papers. Indeed, the normalizing of chronic stress in academia left some researchers unsure as to whether they were at risk of developing, or currently experiencing, difficulties with their health or well-being:

> *". . .even those women who said that they did not experience negative effects on their health due to their academic careers mentioned that they experienced great amounts of stress and contended with sleepless nights, suggesting that those women came to expect extreme stress and lack of sleep as a part of the normal postdoctoral experience"* [54].

Overall, there was a general call for more transparency with regards to managing mental health and well-being in the context of academia:

> *"[A]t high levels there's not very much vulnerability and transparency about how people actually approach their daily work lives and how they actually go about maintaining their wellbeing at the same time as achieving as a researcher"* (Doctoral researcher, Science, UK–[31]).

Due to feelings of uncertainty, and varying levels of mental health literacy, doctoral researchers highlighted the key role of the supervisor in legitimising and aiding in help seeking for mental health or well-being related difficulties:

*"...it really saved me... they weren't going to be my therapist, of course...but they were there to make sure that I addressed my issues"* (Doctoral researcher, Arts & Humanities, UK—[31]).

The lack of open discourse around mental health and well-being related difficulties seemed to perpetuate the idea that a successful academic is infallible and immune to such difficulties. However, this is often not the case, and a large majority of researchers across countries, disciplines and career stages described experiencing ***stress and the presence of physical and mental health difficulties***:

*"...she's got all these publications and she's had grants– ...actually my life is a bloody nervous wreck"* (Professor, Music, Australia—[34]).

*"I suffered severe pain and unknown skin irritations and allergy symptoms. The doctor said everything was caused by stress..."* (Participant, Canada—[53]).

There were some exceptions in the research. For example, when managed using personal resources, the presence of stress was sometimes seen as a motivational factor, and necessary for scholarly development:

*'...seeing stress as "a motivation by itself" urges one to "try harder" and "become more competent and more efficient"...'* (Doctoral researcher, Education, Finland—[42]).

**Coping and support.** ***Support provided by organisations (or lack thereof)*** was touched upon across many of the included papers. Overall, there appeared to be a disconnect between the high expectations set by the higher education system, and the time, resources, and encouragement given to researchers in order to reach these expectations:

*"Just when most academics are due for a break, right when most universities shut down and take offline all of their support services, RGMS [online application process] opens up"* (Senior researcher, Australia—[6]).

Interestingly, UK doctoral researchers [31] perceived that universities prioritized their reputation above acknowledging and addressing their mental health and well-being. Early career researchers across Canada, Germany and the USA also expressed frustration over a lack of change in policy or practice, despite universities publicly acknowledging the biases and challenges faced by under-represented groups in academia, including women and those from a Black ethnic background:

*"...Structural barriers... are documented and real, and yet the universities still have this gender bias problem"* (Post-doctoral researcher—[54]).

There were varying levels of awareness as to what institutional support was currently available (either in terms of mental health or professional development). Some researchers appeared to encounter difficulties in accessing the support provided either due to the support

information not being easily accessible [48], or due to their career stage [54]. As one post-doctoral researcher recounted:

> *"My officemate actually was particularly anxious and he called some kind of help line at [the university] looking for support and they denied him anything as a postdoc. They told him if he were a student okay, or faculty okay, but as a postdoc we can't help you...."* (Post-doctoral researcher, Canada—[54]).

Concerns were also raised as to the effectiveness of student services in handling problems unique to doctoral researchers:

> *"My experience with student services was they didn't know what they could do, they'd say 'I'll look into it'. Granted I'm in quite a unique situation right now, they are multiple things going on. They said 'I don't know if we can do anything to help you, I can look into it and get back to you"* (Doctoral researcher, UK—[48]).

However, others did describe university-based support they had found helpful:

> *'the counselling was great. She really helped me'* (Doctoral researcher, UK—[48]).

The lack of support provided by organisations necessitated the use of **individual coping strategies** to counteract the stress of working in academia. Given the lack of control researchers had over many aspects of the job, many researchers focused on both what they could change and regaining a sense of control. The most common coping strategy mentioned was perseverance, however, a mixture of other emotion-focused and more practical coping strategies were also used, such as re-framing the experience from a negative to a positive and seeking professional help.

> *"My stick-to-it-ive nature. . . has kept me and gotten me to the point where I am, and gotten me to the point where I can finish. . ."* (Doctoral researcher, USA—[38]).

Although not an individual coping strategy per se, the passage of time often allowed for the potency of negative emotions to decrease:

> *"I don't care anymore; I've kind of forgotten about it, to be honest. [. . .] At the time, I was very frustrated and irritated. . ."* (Post-doctoral researcher, Sociology, UK—[44]).

**Factors contributing to job satisfaction** including a passion for science, recognition of hard work from peers or institutions, seeing students develop, or a paper being accepted for publication, also aided in attempts to maintain positive well-being at work:

> *'The science gives me the greatest satisfaction . . . the satisfaction of pitching a question, seeing the results come through'* (Senior researcher, Science, Australia–[51]).

**Perceptions of what support should look like** were included across many of the studies in this review. Most prominent among the suggestions which could improve researchers' mental health and well-being at work was a call for organisations to assess *'productivity relative to opportunity'* [51], and clarity regarding promotion and tenure processes so that individuals

can set realistic goals. Researchers at different career stages also commented on the importance of their physical workspaces engendering a sense of belonging and well-being:

*". . .there no space on this campus where. . .five of us can sit down and just yap without the undergrads constantly taking that space. . .we talk so much about informal learning and. . .we don't have a space for PhD students. . .Places need to grow"* (Doctoral researcher, Social sciences, UK—[31]).

Some suggestions for support were specific to particular academic researcher populations. Post-doctoral researchers explicitly called for more developed policies, programs and workshops tailored to furthering their career [47] whilst doctoral researchers called for supervisors to receive training with regards to conducting supervision and also suggested assigning a supervisor(s) based not only on the research topic, but also on the support needs of the doctoral researcher.

**Positions of privilege.** Academics from different career stages felt privileged to be in a position where they had ***the opportunity to contribute to society*** through their research. Nevertheless, feeling morally compelled to give back to society also had the potential to negatively impact upon well-being through contributing to overwork, and tensions could be found between colleagues and social support networks when moral dispositions did not align:

*"Someone I know who got one of the largest grants ever said, 'I don't care if my research has impact–I'm doing this because I'm curious about this' and I just thought that was an appalling waste of tax payer's money. . ."* (Professor, Education, UK—[34]).

Feelings of tension could also arise when the impact on society or participants was not immediate or direct, particularly when the research involves discussing sensitive topics such as trauma:

'*. . .. the promise of the potential for positive change years down the road does little to help a researcher sleep at night'* [41].

Although already touched upon in other key themes (notably 'work-life balance and the academic lifestyle: an incompatibility' and 'the influence of relationships and role models'), the included papers indicated a sense of inequality in academia that should be drawn out more explicitly.

This sense of inequality was particularly prevalent among responses from female researchers and researchers from a Black ethnic background, who are under-represented among the senior levels in academia, and who often described facing, or expecting to face, incidents of harassment, bias, or discrimination. This left them at risk of ***unequal access to resources, support, and opportunities***, and reduced well-being at work:

*". . . I'm the only Black guy in the group. . . and the only one being treated this way. So, you're like, "What*?!*" you know"* (Doctoral researcher, USA—[38]).

These experiences were considered reflective of society at large. Consequently, both institutional as well as broad societal changes were thought to be needed in order to enact change and foster a more supportive and equal research culture:

'. . . *students reported feeling the need to combat stereotypes that seeped from society at large into their engineering and computing programs'* [38].

Ultimately some researchers noted that:

*'there's definitely a boys/girls' club and being part of that group can help your career'* [51].

## Discussion

We aimed to better understand how researchers experience working in academia, and the effect these experiences have on their mental health and well-being. We identified seven key themes as a result of conducting a meta synthesis across 26 papers which met our inclusion criteria.

The seven key themes spanned across the countries, disciplines and career stages mentioned in the included papers, and shed light on factors at an individual, interpersonal, and systemic level which appear to impact the mental health and well-being of the academic researcher population. However, throughout the analysis, we took care to also highlight the nuances in researchers' experiences. Considering both the similarities and differences in experience can have important implications for workplace policy and can help highlight areas where interventions should be targeted.

Job insecurity, a lack of family-friendly policies, inflexible requirements for funding and promotions, and a push for productivity above all else left many researchers stressed and experiencing (or at risk of experiencing), mental and physical health difficulties. These systemic stressors are highlighted across the wider, albeit limited, literature on this topic [8, 16], and it is unsurprising therefore, that suggestions for support and change across the included studies in this review focused on addressing these systemic issues, as opposed to implementing interventions at the individual level. There was a sense across the included studies that it is scientific/academic practice, and the system's concept of what a successful researcher should look like, that needs to change, rather than putting the onus on the individual to cope in this environment.

Nevertheless, recent evidence suggests that many of these systemic issues continue to pervade academic spaces [56]. Indeed, there remains an expectation to meet high academic performance standards, despite the ongoing disruptions the COVID-19 pandemic has caused across researchers' personal and professional lives [57]. The impact of COVID-19 on the higher education system should continue to be monitored, as evidence suggests that the pandemic has both illuminated and exacerbated the risk particular systemic issues highlighted in this review (such as financial and job insecurity, as well as gender and ethnicity) can have on researchers' mental health and well-being [56, 58].

This review has also highlighted the pressure researchers feel to maintain a reputation of being able to cope with the high expectations set in a competitive academic environment. The stigma that appears to exist with regards to experiencing mental health difficulties in academia, coupled with the normalizing of chronic stress, likely prevents researchers from accessing support when needed [24, 59]. Fostering an environment where open discourse around mental health and well-being at work can occur without fear of repercussions, will likely aide in the detection and treatment, and ultimately prevention, of mental health difficulties [59]. Nevertheless, this review has also highlighted a lack of awareness as to what mental health-based support is currently offered by academic institutions, which represents another barrier to accessing support. Institutions need to ensure that any support currently offered is visible and the process of accessing this support is clear and straightforward. Further research is needed with regards to doctoral researchers and post-doctoral researcher's hopes and expectations for

mental health-based support at work, as some of the included papers in this review [48, 54] have highlighted that they may not be able to access or benefit from the institutional support already provided for students or faculty.

The importance of both peer and social networks in maintaining positive well-being is stated throughout the included papers. Interventions at both an individual and systemic level may be needed to ensure that researchers are not forced to choose their academic identity at the expense of other important life roles and activities, which may in turn limit their access to social support networks. Indeed, belonging to multiple social networks has been shown to be protective of mental health and well-being [54, 60].

In the absence of adequate support from academic institutions, evidence has shown that early career researchers in particular have taken the initiative to form their own peer support networks, an example being Scholar Minds in Germany [61], where workshops related to the PhD experience or general information-sharing and collaboration can take place [62, 63]. These networks can help to foster a sense of community and connection [63]. Whilst it is imperative to further develop peer support systems where opportunities exist [64], it is important to note that this review has highlighted that these networks can also be a source of competition and stress. Any peer support interventions will therefore warrant careful evaluation, and care will need to be taken to ensure that these interventions do not overburden an already overburdened workforce.

Despite finding multiple similarities across career stages, disciplines and countries, this review also highlighted some notable differences in experience between certain subgroups of the academic researcher population.

Across the included studies which commented on the experiences of doctoral researchers, the key role of the supervisor was highlighted, a sentiment which is echoed across the wider literature on doctoral researchers' mental health and well-being [24, 65]. As such, it is important for universities to invest in this relationship by providing appropriate training for supervisors, clarifying the supervisory role, and ensuring a good fit between doctoral researcher and supervisor based on both professional and personal support needs. Whilst some universities may already have procedures linked to these suggestions in place [66], this review suggests that the supervisory relationship can still be a source of tension for doctoral researchers. Further research is needed to explore the supervisory relationship from both a doctoral researcher and supervisor perspective [67], to help identify ways to ensure that this relationship does not impact negatively on the well-being of either individual.

This review also highlighted the difficulties faced by female researchers and researchers from a Black ethnic background in particular, although it is important to note that other under-represented groups in academia including those from the LGBTQ+ community and those with disabilities also experience similar systemic challenges with regards to a lack of role models [68] and experiences of bias and discrimination [8, 68].

Initiatives have been introduced to try to tackle inequality in academia across the UK, Europe, North America, and Australia [69, 70]. A notable example is the Athena SWAN Charter, introduced in the UK in 2005, which provides incentives and awards for higher education institutions to actively highlight and tackle gender inequality across multiple disciplines [69]. Whilst participation in these initiatives has been shown to increase awareness of broader diversity issues and has helped to challenge incidents of discrimination and bias [69], the results of this review, with all papers published since 2011, suggest that researchers from under-represented groups in academia still experience the academic research environment as unequal and unsupportive. Indeed, in the short term at least, evidence from the wider literature suggests that some academics may perceive these initiatives as restricted in their ability to tackle persistent pay, power and promotion disparities [69, 71]. Whilst calls for societal change inherent in

this review and the wider literature [70] may be beyond the scope of the higher education system, further research could explore in greater depth the work experiences of those from under-represented groups, along with their perspectives on what more effective support could look like.

Strengths and limitations of the included papers

Most of the articles reviewed were judged to be of good quality, and each article shed light on how working in academia can impact on researchers' well-being and mental health. Nevertheless, there are a number of limitations inherent in the papers included in this review. First, there was a notable lack of reflection into how the researchers themselves may have influenced the findings of their studies. Through not clearly stating the studies philosophical stance nor clearly stating the potential impact of the researchers, it is difficult to ascertain how the research teams' characteristics may have influenced the process of data collection and analysis. Second, the link between work experiences and researchers' mental health and wellbeing was not always explicitly stated in the papers, and it therefore fell to us as the reader to make our own interpretations and inferences regarding the data—which may have been different from the study participant's/author's original intended point. Indeed, due to the differences in how mental health and well-being were conceptualised and discussed across the included papers, it was difficult to maintain a distinction between the two concepts when conducting our analysis. Finally, this review has also highlighted the general scarcity of research which explores academic researchers' mental health and well-being experiences–only a small number of articles were identified and included in this review (n = 26).

## Strengths and limitations of the meta-synthesis

The meta-synthesis itself also has some notable limitations. The search strategy involved an English language restriction, therefore the majority of included papers included participants from predominantly Western, English-speaking countries. As such, the findings from this review may not reflect the views and experiences of researchers working in higher education institutions across the globe. However, research exploring the stressors faced by academic researchers suggests that there are similarities between the experiences of those in western countries and the rest of the world, particularly with regards to unequal access to resources, support, and opportunities [72]. Our search strategy took a broad approach, as we aimed to provide an inclusive and in-depth examination of the status of academic researchers' mental health and well-being. As a result, we included a range of academic researcher populations, methodological approaches, constructs related to mental health and well-being, and places/institutions of higher education. Nevertheless, a narrower search strategy might have allowed for a more in-depth look into more specific practices and experiences and could be a potential avenue for further systematic research in this area.

On a similar note, well-being and mental health are complex constructs and we acknowledge that the list of terms related to these constructs that we included in the search strategy to help identify relevant papers was by no means exhaustive. By not including a more exhaustive list of constructs related to mental health and well-being, we may have missed further relevant papers. Nevertheless, to help ensure that we captured as many relevant papers as possible, we conducted a preliminary informal literature search to discover how the existing relevant literature conceptualised these constructs.

It is also important to note that many of the papers included doctoral researchers as participants in their study (n = 14), and therefore their views may arguably be more prevalent and represented in this meta-synthesis than other academic researcher groups. A relatively small number (n = 12) of the included papers in this review focused specifically on one academic

researcher group past PhD level, highlighting a dearth of exploratory research into the well-being and support needs of post-doctoral researchers and those in the later stages of their career which, again, may form an avenue for future research. It should also be noted that the views of researchers who have experienced difficulties with their mental health or well-being may arguably be more prevalent across the included papers (and thus, more represented in our analysis), as these experiences may make them more inclined to participate in research exploring these concepts [48]. Similarly, some of the included papers specifically sought out participants with symptoms of a mental health difficulty [39], and others specifically focused on examining more negative constructs which could contribute to poorer mental health and well-being, such as stress [49].

Finally, whilst the impact of researching trauma on mental health was noted explicitly [41], as was the negative impact of the 'impact agenda' on researchers from less applied, theoretical disciplines [34], any other experiences specific to discipline/subject area and the impact these have on mental health and well-being were difficult to ascertain, given that disciplines were not always stated, and ascertaining disciplinary differences was often not the focus of the included papers.

## Conclusion

The findings of this systematic review and qualitative meta-synthesis highlight the individual, interpersonal, and systemic factors that can impact the mental health and well-being of researchers who work in academia. Attempts to navigate the high expectations set by the academic system, continued job insecurity, and incidents of bias and discrimination have left researchers experiencing, or at risk of experiencing, physical and mental health difficulties. This review has highlighted areas where better support could be implemented, including maximising opportunities for social and peer support, and tackling systemic issues. Further high-quality qualitative research is needed to better understand how systemic change, including tackling inequality, can be brought about more immediately and effectively from a researcher's perspective. Further high-quality qualitative research is also needed to better understand the experiences and support needs of post-doctoral and more senior researchers as there is a paucity of literature in this area.

## Supporting information

**S1 File. PRISMA checklist.**
(PDF)

**S2 File. Full search strategy.**
(PDF)

## Acknowledgments

We would like to thank Dr. Vanessa Pinfold, research director and co-founder of The McPin Foundation, for their helpful comments on the final draft of this manuscript.

## Author Contributions

**Conceptualization:** Helen Nicholls, Danielle Lamb, Jo Billings.

**Formal analysis:** Helen Nicholls, Matthew Nicholls, Sahra Tekin, Danielle Lamb, Jo Billings.

**Funding acquisition:** Jo Billings.

**Investigation:** Helen Nicholls, Matthew Nicholls, Sahra Tekin.

**Methodology:** Helen Nicholls, Matthew Nicholls, Danielle Lamb, Jo Billings.

**Resources:** Helen Nicholls.

**Supervision:** Danielle Lamb, Jo Billings.

**Visualization:** Helen Nicholls.

**Writing – original draft:** Helen Nicholls.

**Writing – review & editing:** Helen Nicholls, Sahra Tekin, Danielle Lamb, Jo Billings.

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
