## [Decision Letter · Decision Letter 0]

6 Apr 2022

PONE-D-22-03203The impact of working in academia on researchers' mental health and well-being: A systematic review and qualitative meta-synthesisPLOS ONE

Dear Dr. Nicholls,

Thank you for submitting your manuscript to PLOS ONE. After careful consideration, we feel that it has merit but does not fully meet PLOS ONE’s publication criteria as it currently stands. Therefore, we invite you to submit a revised version of the manuscript that addresses the points raised during the review process.

 This is a well-written, and methodologically sound meta-synthesis of the literature on the relationship between academia and mental health/well-being. In addition to the reviewer comments noted below, it would be useful for the authors to include a brief overview in the Introduction of what quantitative-focused studies in this area have found. There are many validated mental health and well-being scales which provide a empirical sense of these issues, and for readers unfamiliar with this area, understanding the prevalence and severity of mental ill-health among academics would be useful. The authors could then better highlight the current gaps in understanding about this problem given limitations of quantitative-focused studies as way to strengthen their case for only focusing on qualitative studies.   

We look forward to receiving your revised manuscript.

Kind regards,

Michelle Torok, Ph.D.

Academic Editor

PLOS ONE

Journal Requirements:

3. Please upload a copy of Figure 1, to which you refer in your text on page 10. If the figure is no longer to be included as part of the submission please remove all reference to it within the text.

Reviewers' comments:

Reviewer's Responses to Questions

**Comments to the Author**

1. Is the manuscript technically sound, and do the data support the conclusions?

Reviewer #1: Yes

Reviewer #2: Yes

2. Has the statistical analysis been performed appropriately and rigorously? 

Reviewer #1: N/A

Reviewer #2: N/A

3. Have the authors made all data underlying the findings in their manuscript fully available?

Reviewer #1: Yes

Reviewer #2: Yes

4. Is the manuscript presented in an intelligible fashion and written in standard English?

Reviewer #1: Yes

Reviewer #2: Yes

5. Review Comments to the Author

Reviewer #1: Introduction

The authors explain the relevance of the review by referring to the increased demands on academics, limited literature on mental health and wellbeing in academia and heterogeneity of the groups working in academia, as well as of the constructs of mental health and wellbeing. These introductory paragraphs are well written and provide a clear understanding to the readers on why such a review is needed. However, the authors do not explain what they mean by mental health and wellbeing, the potential overlap between the terms or the clear differences. This is attempted in the methods section (lines 137-140); however, these are mostly examples, and the review would benefit from a more thorough definition of the two concepts from the start. Furthermore, the introduction lacks a clear rationale as of why the authors have chosen to synthesise qualitative published data only. I can see that this is briefly mentioned in the methods section (lines 151-152), but a clear rationale from the beginning would help the readers.

Methods

- Line 141 – I think it would be better to clarify criterion (a) – how did the authors determine if the articles focused sufficiently on researchers’ mental health and well-being experiences? What counted as “sufficient”? Did each article have to investigate both constructs, or was one or the other sufficient for inclusion?

- Lines 170-171 – how did the authors decide that these eight papers were the “most immediately relevant and current papers”?

- Line 187 – though the diversity of the research team is a clear strength of the review, and it is great to see that the different backgrounds and expertise of each researcher is discussed, I think it might be an overstatement to say that this ensured that any personal assumptions or blind spots were identified – perhaps this expertise helped to minimise such aspects

Results

- Well written

- Line 394 – I think the authors meant “affecting”?

Discussion

Line 646 – While the work here is extremely important and a great start in synthesising researchers’ experiences across a number of geographical regions, I think it is important to acknowledge that the representativeness of these experiences is still under question, especially as there were no papers from Africa, one from Asia (only China) and most papers were from developed regions of the world (North America, Europe, Australia and Oceania). Therefore, the authors must be careful when stating that these themes “transcended geographical…boundaries”

Reviewer #2: In their systematic review and qualitative meta-synthesis, the Authors explore the literature identifying individual and systemic factors related to the academic work and point out how these factors impact mental health and well-being of academic researchers. Spanning several countries, career stages, and foci, they provide a detailed analysis of the identified 7 major factors. This is important work with a potential to better our understanding of the complex relationships between the academic culture, system, work conditions, and scholars’ mental health and well-being. While I cannot judge the methods and their execution in the study (I do not have experience with qualitative reviews), I have some suggestions and comments which could further improve the manuscript. Please see below.

1. Some working definitions of the main concept in the study should be explicitly stated in the introduction. These are: mental health and wellbeing (plus the difference); early career researchers (ECRs); the difference between “mental health” and “mental health experiences” (does it mean self-reported mental health in comparison to clinical screening tools?). The last is particularly important for those readers who are more familiar with quantitative literature.

2. I’d suggest using the term “doctoral researchers” more than “doctoral students”. PhD is a training and a job, but it’s very different from studying on the BA or MA levels. The term “doctoral researchers” seems to align better with the effort to mitigate too dominant power dynamics in academia

3. As the Authors mention, there are some initiatives aiming to incentivise improvements in academia (like Athena SWAN Charter), however, for the sake of completeness, it may be valuable to mention that there are many bottom-up initiatives organised and maintained by ECRs which aim to both describe and improve the situation. It is meaningful that ECRs often take the initiative to help each other while missing enough support from their institutions. It also speaks to the need for peer support, emphasised in the article. I can only make some suggestions from Germany (http://scholarminds.net, https://www.phdnet.mpg.de/n2), but I am sure there are more out there in Europe and beyond.

4. Figure 2: It’s a very informative figure, but the research question in the middle of the figure doesn’t fit there. Maybe “main themes in mental health/well-being research” or something similar would be more adequate?

5. Lines 74-79: these two sentences don’t seem to follow one another. Did the Authors mean that secure employment, autonomy, and teamwork are less present in the life of PhD students?

6. One thing that is generally not clear is how prevalent the issues are. The Authors should mention the problem with biased data/convenience sample: those who are experiencing mental health/wellbeing issues are more likely to participate in studies investigating these topics, especially when the studies are qualitative (which normally require more time than, for example, forced-choice surveys). One way to discuss this would be to look into recruitment strategies in the analysed papers. Nevertheless, the bias could not be fully excluded that way, so a comment should be made.

7. There is an emphasis on the distinction between mental health and well-being in line 137-140, which is useful given that these two may require different strategies, responsible bodies, time frames, legal structures, and staff to improve the issues. However, this distinction is blurred in the results and neglected in the discussion. If this was done because it is simply hard to point out the differences in the data, it should be mentioned. If the reason was that the 7 identified factors seem to influence both in the same or at least similar way, it is also an interesting comment and should be mentioned.

8. Some statements in the text seem a little strong. However probable, these statements may give a false impression of an existing causal relationship which may be an (possibly correct!) interpretation of the Authors (as they admit themselves in the discussion of limitations). If the Authors can elaborate on it further (maybe with more examples), that would be beneficial. If not, perhaps a weaker version should be considered (and maybe suggested for future research to follow up on). Some examples:

o In lines 373-374: “Due to this culture of silence however, there was a tendency for researchers to blame any difficulties or dissatisfaction associated with their job on themselves”. However probable, it seems like a rather strong statement between “the culture of silence” and self-blame.

o In lines 392-395: “Ultimately, being perceived as meeting or not meeting the expectations (…) can have an impact on a researchers’ confidence in their ability to do their job, not only effecting their well-being, but also their sense of identity as an academic, and thus their sense of belonging to the academic community.”

9. Just a comment: I applaud the transparency achieved with the “Reflexivity” part of the manuscript and the discussion of researchers’ influence in “Strengths and limitations of the included papers”.

10. There is a growing number of quantitative works using screening tools and custom questionnaires to provide evidence for elevated levels of depression, anxiety, burnout, etc. in ECRs. Even though this is a qualitative analysis, the authors insist on scarcity of research done in the field of mental health of academics without citing many of works that have directly estimated the scope of the problem. If only for the sake of dissemination and relatedness between studies using different methods but on the same topic, it should be at least noted that they exist. To name a few:

o Evans, T. M., Bira, L., Gastelum, J. B., Weiss, L. T., & Vanderford, N. L. (2018). Evidence for a mental health crisis in graduate education. Nature Biotechnology, 36 (3), 282–284. https://doi.org/10.1038/nbt.4089

o Satinsky, E. N., Kimura, T., Kiang, M. V., Abebe, R., Cunningham, S., Lee, H., ... Tsai, A. C. (2021). Systematic review and meta-analysis of depression, anxiety, and suicidal ideation among Ph.D. students. Scientific Reports, 11 (1), 1–12. https://doi.org/10.1038/s41598-021-93687-7

o Woolston, C. (2017). Graduate survey: A love–hurt relationship. Nature, 550 (7677), 549–552. https://doi.org/10.1038/nj7677-549a

11. After all the work the Authors put into this analysis, and given their insightful conclusions, as a reader I remain wanting to learn more about their own suggestions for how to improve the situation in academia. They appear here and there in the discussion, yet, even if only speculative and brief, the Authors could use their acquired insights in the topic to clearly name some evident candidates for a change. However, this is a personal choice of the Authors and leaving the several suggestions only embedded in the discussion would not make the paper lacking.

12. Two tiny things unrelated to the content: There’s a typo in line 93 (aide instead of aid) and there are some misplaced or missing quotation marks, apostrophes, and brackets throughout the text.

6. PLOS authors have the option to publish the peer review history of their article (what does this mean?). If published, this will include your full peer review and any attached files.

Reviewer #1: No

Reviewer #2: **Yes: **Magdalena Matyjek

---

## [Author Response · Author response to Decision Letter 0]

14 Apr 2022

We would like to thank both the academic editor and the reviewers for their valuable and insightful comments on our manuscript. Please find our point-by-point response to both reviewer and editor comments in our response letter attached to this submission.

---

## [Decision Letter · Decision Letter 1]

2 May 2022

PONE-D-22-03203R1The impact of working in academia on researchers' mental health and well-being: A systematic review and qualitative meta-synthesisPLOS ONE

Dear Dr. Nicholls,

Thank you for submitting your manuscript to PLOS ONE. After careful consideration, we feel that it has merit but does not fully meet PLOS ONE’s publication criteria as it currently stands. Therefore, we invite you to submit a revised version of the manuscript that addresses the points raised during the review process.

We look forward to receiving your revised manuscript.

Kind regards,

Giuseppe Filiberto Serraino, M.D., Ph.D.

Academic Editor

PLOS ONE

Journal Requirements:

Reviewers' comments:

Reviewer's Responses to Questions

**Comments to the Author**

1. If the authors have adequately addressed your comments raised in a previous round of review and you feel that this manuscript is now acceptable for publication, you may indicate that here to bypass the “Comments to the Author” section, enter your conflict of interest statement in the “Confidential to Editor” section, and submit your "Accept" recommendation.

Reviewer #1: All comments have been addressed

Reviewer #2: (No Response)

2. Is the manuscript technically sound, and do the data support the conclusions?

Reviewer #1: Yes

Reviewer #2: Yes

3. Has the statistical analysis been performed appropriately and rigorously? 

Reviewer #1: Yes

Reviewer #2: N/A

4. Have the authors made all data underlying the findings in their manuscript fully available?

Reviewer #1: Yes

Reviewer #2: Yes

5. Is the manuscript presented in an intelligible fashion and written in standard English?

Reviewer #1: Yes

Reviewer #2: Yes

6. Review Comments to the Author

Reviewer #1: (No Response)

Reviewer #2: Thank you for your revisions. I believe this is important work which should be shared with other researchers. A tiny comment - while reading the revised version, I came across a typo in line 93: poor work like balance (instead of life). Congratulations for the authors on the work and good luck for future projects.

7. PLOS authors have the option to publish the peer review history of their article (what does this mean?). If published, this will include your full peer review and any attached files.

Reviewer #1: No

Reviewer #2: **Yes: **Magdalena Matyjek

---

## [Author Response · Author response to Decision Letter 1]

5 May 2022

We would like to thank the academic editors and reviewers who have reviewed our manuscript. Please find our response to the comments made in the attachment 'Response to Reviewers'.

---

## [Editor Report · Decision Letter 2]

11 May 2022

The impact of working in academia on researchers' mental health and well-being: A systematic review and qualitative meta-synthesis

PONE-D-22-03203R2

Dear Dr. Nicholls,

We’re pleased to inform you that your manuscript has been judged scientifically suitable for publication and will be formally accepted for publication once it meets all outstanding technical requirements.

Kind regards,

Giuseppe Filiberto Serraino, M.D., Ph.D.

Academic Editor

PLOS ONE

---

## [Editor Report · Acceptance letter]

16 May 2022

PONE-D-22-03203R2 

The impact of working in academia on researchers' mental health and well-being: A systematic review and qualitative meta-synthesis 

Dear Dr. Nicholls:

I'm pleased to inform you that your manuscript has been deemed suitable for publication in PLOS ONE. Congratulations! Your manuscript is now with our production department. 

Kind regards, 

on behalf of

Professor Giuseppe Filiberto Serraino 

Academic Editor

PLOS ONE